# Self-organization of primitive metabolic cycles due to non-reciprocal interactions

Vincent Ouazan-Reboul[1], Jaime Agudo-Canalejo [1] & Ramin Golestanian [1,2] ✉

One of the greatest mysteries concerning the origin of life is how it has emerged so quickly after the formation of the earth. In particular, it is not understood how metabolic cycles, which power the non-equilibrium activity of cells, have come into existence in the first instances. While it is generally expected that non-equilibrium conditions would have been necessary for the formation of primitive metabolic structures, the focus has so far been on externally imposed non-equilibrium conditions, such as temperature or proton gradients. Here, we propose an alternative paradigm in which naturally occurring non-reciprocal interactions between catalysts that can partner together in a cyclic reaction lead to their recruitment into self-organized functional structures. We uncover different classes of self-organized cycles that form through exponentially rapid coarsening processes, depending on the parity of the cycle and the nature of the interaction motifs, which are all generic but have readily tuneable features.

Since Oparin[1] proposed a picture to describe how early forms of living matter might have emerged from what Haldane described as the prebiotic soup[2], there has been a significant amount of progress in our understanding of the physical aspects of the origin of life[3]. Recent examples of such studies include spontaneous emergence of catalytic cycles[4,5], spontaneous growth and division of chemically active droplets[6–9], programmable self-organization of functional structures under non-equilibrium conditions[10,11], and controllable realization of metabolically active condensates[12]. A striking generic observation that has emerged in a variety of different scenarios is that the introduction of non-equilibrium activity in the form of catalytic activity, or a primitive form of metabolism, can be a versatile driving force for functional structure formation[13–18] with manifestations of lifelike behaviour[19–27]. It has also been demonstrated that the structured catalytic activity that would support the required non-equilibrium processes for primitive cells can be successfully coupled with the condensation of appropriate functional nucleotide and peptide components in membrane-free systems[28–30], as well as lipid components in protocells with functionalized membranes[31,32].

Living systems necessarily involve a set of auto-catalytic chemical reactions[33], which have been theoretically shown to spontaneously emerge in a population of polypeptide-like structures that could assemble in a primordial soup setting[34–37]. A candidate metabolic cycle that may have served a key role in the early stages of life formation is the citric acid cycle, which consists of 11 members and exhibits evolutionary robustness and universality[38,39]. Candidates for pre-RNA and protein autocatalytic chemical networks have been identified from early microbial organisms[40], and mixtures of RNA fragments have been experimentally observed to organize into self-replicating and catalyzing reaction networks[41–44].

The physicochemically motivated ideas initiated by Oparin and Haldane were critically debated for much of the past century by proponents of the perspective that (genetic) information should be considered as the main organizer of matter that forms life[33,45,46]. As a modern interpretation of these considerations, we note that the currently accepted paradigm assumes that the ingredients that would later join up to form intricate components of living systems first come together by ad hoc physical forces without any input from the *information* that will eventually be at work in their hierarchical self-organization. The information-based organization is expected to occur when the system has already made physical condensates. However, this paradigm has so far been unable to answer two important questions. First, polymeric condensates such as coacervates are known to be intrinsically very slow, almost glass-like, in their dynamics, even if

[1]Max Planck Institute for Dynamics and Self-Organization, Am Fassberg 17, D-37077 Göttingen, Germany. [2]Rudolf Peierls Centre for Theoretical Physics, University of Oxford, OX1 3PU Oxford, UK. ✉e-mail: ramin.golestanian@ds.mpg.de

they are driven by external non-equilibrium forces like temperature and proton gradients. Therefore, it is not clear how such dense and glassy condensates that were formed randomly would have been able to efficiently evolve to form information-based functional structures via random searches given the time scale that has taken life to emerge after the formation of the earth. Secondly, the physics of condensation is governed by relatively slow power law coarsening dynamics such as the Lifshitz-Slyozov (~ $t^{1/3}$) law[47], even in externally driven non-equilibrium cases. Then, it is unclear from a physical point of view how life has emerged through slow coarsening into inherently slow condensates.

In connection with the above considerations and to broaden the scope of the research on the physical aspects of the origin of life, we pose the following question: how can we envisage pathways in which the information contained in chemical reaction networks from which primitive forms of metabolism can emerge would lead to structural self-organization of the corresponding components? Here, we propose a strategy that can achieve this task by employing the naturally occurring non-reciprocal interactions between catalysts that can form a cyclic reaction network. We show that model catalytically-active particles participating in a metabolic cycle are able to spontaneously self-organize into condensates, which may aggregate or separate depending on the number of particle species involved in the cycle, and exhibit chasing, periodic aggregation and dispersal, as well as self-stirring, thus providing a generic mechanism for spontaneous formation of metabolically-active protocells. While the observed

(super-)exponential coarsening law offers a significantly faster alternative for the formation of condensates (see Fig. 1), the information-driven dynamics leads to formation of structurally active and functional condensates that exhibit lifelike behaviour already at the outset.

## Results

Non-reciprocal interactions have been shown to generically emerge in active matter in the context of non-equilibrium phoretic (chemotactic) interactions[48]. Chemotaxis in response to chemical gradients has been experimentally observed not only in the context of synthetic micro-scale colloids, but also for biological enzymes[49,50], molecular catalysts[51], and nucleic acids[52,53]. The latter two observations raise the interesting prospect that chemotactic interactions may have played a role in the assembly of prebiotic systems, e.g. in an RNA world scenario. Let us consider a set of $M$ species of chemically-active particles (Fig. 1a, top), representing catalyst molecules or enzymes. Each of the particles converts a substrate (s) into a product (p) at a rate $\alpha$. At steady state, they create perturbations in the concentration field of the corresponding substrate that decays with distance $r$ as $\delta c^{(s)} \propto -\alpha/r$, and a corresponding change in the concentration of the corresponding product as $\delta c^{(p)} \propto \alpha/r$ (Methods). These particles are also chemotactic (Fig. 1a, bottom): when subjected to a concentration gradient of their substrate, they develop a velocity $v \propto -\mu^{(s)} \nabla c^{(s)}$ with $\mu^{(s)}$ the chemotactic mobility for the substrate, which is negative or positive if the particle is attracted to or repelled from the substrate, respectively.

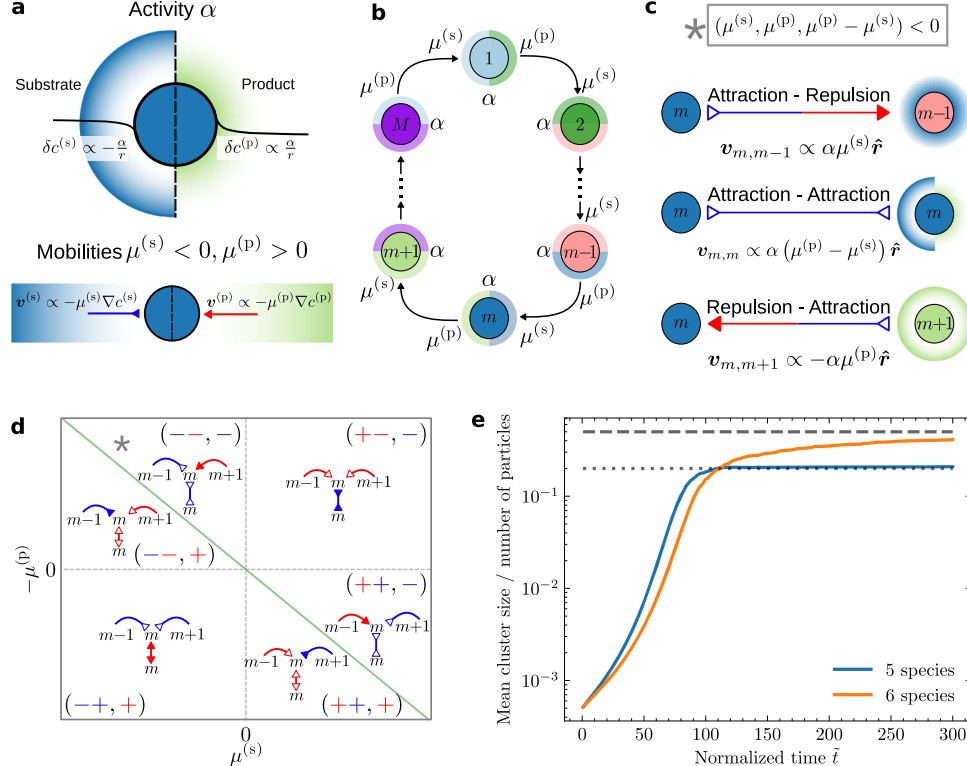

**Fig. 1 | Properties and interactions of catalytically-active particles. a** The particles convert substrate (s) into product (p) with a rate given by the activity $\alpha$ (top) and respond to gradients of these chemicals with mobilities $\mu^{(s)}$ and $\mu^{(p)}$ (bottom). **b** $M$ active particle species are arranged in a model metabolic cycle, in which the product of species $m$ is the substrate of species $m+1$. **c** Non-reciprocal interactions between particles of the same ($v_{m,m}$) or adjacent ($v_{m,m\pm1}$) species. Direction and colour of arrows indicate the attractive (blue, inwards arrowhead) or repulsive (red, outwards arrowhead) nature of the interaction. $\hat{r}$ is the unit vector pointing from particle $m$ to the other particle. **d** Phase diagram of interaction motifs. Each region constrains the mobilities so that one interaction has a higher magnitude than the

others, as highlighted by a full arrowhead. The grey asterisk indicates the location in parameter space of the interactions pictured in (**c**). The green line separates the self-attracting and self-repelling regions. The sign triplets correspond to the signs of ($\mu^{(s)}, \mu^{(p)}, \mu^{(p)} - \mu^{(s)}$). **e**, Cluster growth dynamics for a cycle of $M=5$ (blue, see Fig. 3a and Supplementary Movie 7) and $M=6$ (orange, see Fig. 3b and Supplementary Movie 3) species, showing super-exponential coarsening dynamics of the condensate formation. For $M=5$ species, the mean cluster size saturates at a value corresponding to the total particle population divided by $M$ (grey dotted line). For $M=6$, it saturates at half the particle population (dashed grey line).

Similarly, the particles are able to chemotax in response to gradients of their products, with a mobility $\mu^{(p)}$.

To create a model for primitive metabolism, we consider a simplified metabolic cycle (Fig. 1b), in which the substrate of the catalyst species $m$, which we denote as chemical $(m)$, is the product of species $m-1$. To close the cycle, species 1 has the product of species $M$ as its substrate. For simplicity, we take all catalyst species to have the same parameters $\alpha, \mu^{(s)}$ and $\mu^{(p)}$, and to be present in the system at identical initial concentrations. This assumption will not limit the range of validity of the predictions, as more general choices for the parameters can be shown to lead to a number of distinct classes with relatively sizeable regions of the parameter space for each behaviour[54,55]. The cycle can achieve a steady state without net chemical production or consumption. Due to their chemical activity and chemotactic mobilities, the particle species can interact with one another through chemical fields (Fig. 1c). For instance, if we consider two particles of species $m$ and $m-1$, then the particle of species $m-1$ creates, through its chemical activity, a concentration gradient of the substrate of the particle of species $m$, to which the latter responds by developing a velocity directed towards the particle of species $m-1$, $\boldsymbol{v}_{m,m-1} \propto \alpha\mu^{(s)}\hat{\boldsymbol{r}}$, where $\hat{\boldsymbol{r}}$ is the unit vector pointing from the particle that creates the perturbation to the particle that responds to the perturbation (Methods). On the other hand, the particle of species $m$ consumes the product of $m-1$, and thus the particle of species $m-1$ develops a velocity $\boldsymbol{v}_{m-1,m} \propto -\alpha\mu^{(p)}\hat{\boldsymbol{r}}$ towards the other particle. As a consequence, the interactions between the particles of species $m$ and $m-1$ are nonreciprocal, i.e. $\boldsymbol{v}_{m,m-1} \neq -\boldsymbol{v}_{m-1,m}$ (see Fig. 1d for different possibilities). This effective violation of action-reaction symmetry is a signature of non-equilibrium activity, leading to non-trivial many-body behaviour as has been shown for chemically-active particles interacting through a single chemical[25,56], active mixtures interacting through generic short-range interactions[57,58], complex plasmas[59], and other systems[60,61]. Particles of the same species also self-interact by consumption of their substrate and creation of their product, with a velocity $\boldsymbol{v}_{m,m} \propto \alpha(\mu^{(p)} - \mu^{(s)})\hat{\boldsymbol{r}}$. We note that these effective non-reciprocal interactions mediated by chemical fields are long-ranged, with the induced velocities going as $1/r^2$ (Methods).

We consider the evolution equations for the concentration fields of the active species $\rho_m$ and their substrates $c^{(m)}$, given by the coupled system of $2M$ equations

$$\partial_t \rho_m(\boldsymbol{r}, t) = \nabla \cdot [D_p \nabla \rho_m + (\mu^{(s)} \nabla c^{(m)} + \mu^{(p)} \nabla c^{(m+1)})\rho_m], \tag{1}$$

$$\partial_t c^{(m)}(\boldsymbol{r}, t) = D\nabla^2 c^{(m)} + \alpha(\rho_{m-1} - \rho_m). \tag{2}$$

Equation (1) describes the conserved dynamics of the catalysts, with a diffusion term involving a species-independent coefficient $D_p$ and a chemotactic drift term in response to both substrate and product gradients. The substrate concentrations evolve according to the reaction-diffusion Eq. (2), with a diffusion coefficient $D$, and a reaction term corresponding to the activity of the catalysts.

The time evolution of Eqs. (1) and (2) naturally leads to the formation of clusters, akin to active phase separation[25,56]. The clusters are formed through a particularly fast and efficient coarsening process that exhibits exponential growth rather than the commonly occurring power law form, associated with processes such as Ostwald ripening, as can be seen in Fig. 1e (see Methods). This behaviour can be characterized using a simple scaling argument. When particles are collapsing onto a cluster, the rate of growth for the cluster can be estimated as $\frac{dN}{dt} = \oint_S \rho \boldsymbol{v} \cdot d\boldsymbol{S}$ where the velocity $\boldsymbol{v} = -\mu\nabla c$ can be expressed in terms of the particle concentration by using Gauss theorem and the relation $-\nabla^2 c = \alpha\rho/D$, which yields $\frac{dN}{dt} = \frac{\mu\alpha}{D}\rho N$. This expression can be integrated to obtain $N(t) = N_0 \exp\left(\frac{\mu\alpha}{D}\int_0^t dt_1\rho\right) \simeq N_0 \exp\left(\frac{\mu\alpha}{D}\rho t\right)$, which

predicts an exponential growth law for constant $\rho$ and allows for super-exponential growth if the density increases with time, which matches well with the results presented in Fig. 1e.

This observation suggests that non-equilibrium phoretic interactions have the ability to guide formation of dense clusters in a fast and efficient manner, and as such, can be strong candidates for creating the appropriate conditions for the emergence of early functionalized protocells.

A linear stability analysis on Eqs. (1) and (2) (Methods) around a spatially-homogeneous solution leads to the following eigenvalue equation for the macroscopic (long-wavelength) particle density modes

$$-\sum_{n=1}^{M} \Lambda_{m,n} \delta\rho_n = \lambda\delta\rho_m. \tag{3}$$

The matrix $\Lambda_{m,n}$ describes the velocity response of species $m$ to species $n$, and is defined as follows

$$\begin{cases} \Lambda_{m,m-1} = \alpha\mu^{(s)}\rho_0/D, \\ \Lambda_{m,m} = \alpha(\mu^{(p)} - \mu^{(s)})\rho_0/D, \\ \Lambda_{m,m+1} = -\alpha\mu^{(p)}\rho_0/D, \\ \Lambda_{m,n\neq\{m,m\pm1\}} = 0, \end{cases} \tag{4}$$

where $\rho_0$ represents the initial homogeneous concentrations. By definition, $\Lambda_{m,n}$ is negative, or positive, if $m$ is attracted to, or repelled from, $n$, respectively. The form of $\Lambda_{m,n}$ suggests a classification scheme as there are six possible interaction motifs (Fig. 1d), representing the interactions of each species with itself as well as with its two neighbours in the metabolic cycle. The signs of the interactions are represented diagrammatically, following the conventions defined in Fig. 1c and d.

The eigenvalues $\lambda_\ell$ ($\ell \in \{1, ..., M\}$) allow us to predict the stability of the system: $\text{Re}(\lambda) > 0$ for any eigenvalue $\lambda$ indicates an instability, whereas $\text{Re}(\lambda) < 0$ for all eigenvalues implies a stable homogeneous state. The eigenvector $\delta\rho_m^\ell$, in turn, gives the stoichiometry at the onset of instability, i.e. the ratio of the different species within the growing perturbation, which may be positive, for species that aggregate together, or negative, for species that separate.

The topology of the metabolic cycle strongly influences its self-organization. As a point of comparison, we consider a non-cyclic system, in which $M$ catalytic species all act on a single chemical field. In this case, all the coefficients of the interaction matrix are equal to $\alpha\mu\rho_0$, leading to a system with only one nonzero eigenvalue $\lambda = -M\alpha\mu\rho_0/D$. The corresponding instability condition is $\alpha\mu < 0$, and the instability is equivalent to that of the Keller-Segel model[25,62]. The model metabolic cycle studied here, however, presents a different category. As the interaction matrix (4) is a circulant matrix, its eigenvalues are easily calculated as

$$\begin{cases} \text{Re}(\lambda_\ell) = -\frac{\alpha\rho_0}{D}(\mu^{(p)} - \mu^{(s)})[1 - \cos(2\pi\ell/M)], \\ \text{Im}(\lambda_\ell) = \frac{\alpha\rho_0}{D}(\mu^{(s)} + \mu^{(p)})\sin(2\pi\ell/M), \end{cases} \tag{5}$$

(see Supplementary Note 1 for graphical representations of the eigenvalue spectra for different species numbers). There are now $M-1$ nonzero eigenvalues, which come as pairs of complex conjugate numbers with the possible exception of $\lambda_{M/2}$ for $M$ even. In stark contrast with the non-cyclic system, the complex character of these eigenvalues opens the door to oscillatory behaviour. The instability condition, obtained by imposing that the real part of at least one eigenvalue is larger than zero, in turn corresponds to

$$\mu^{(p)} - \mu^{(s)} < 0, \tag{6}$$

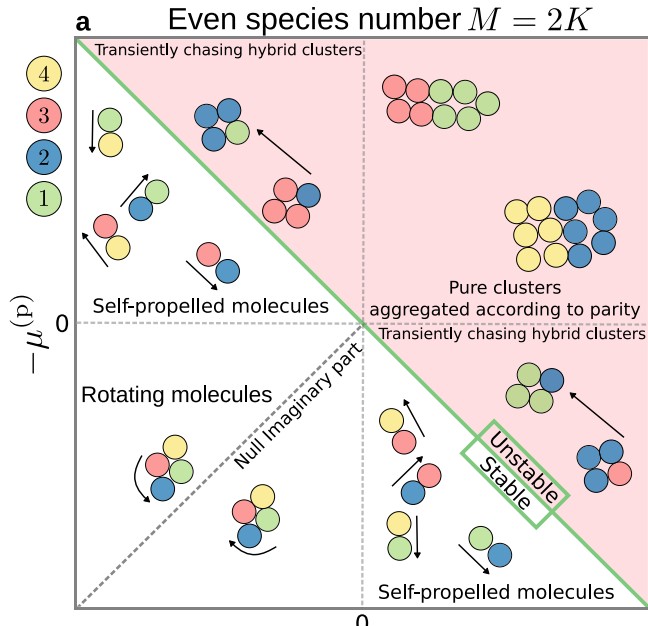

**a**  Even species number $M = 2K$

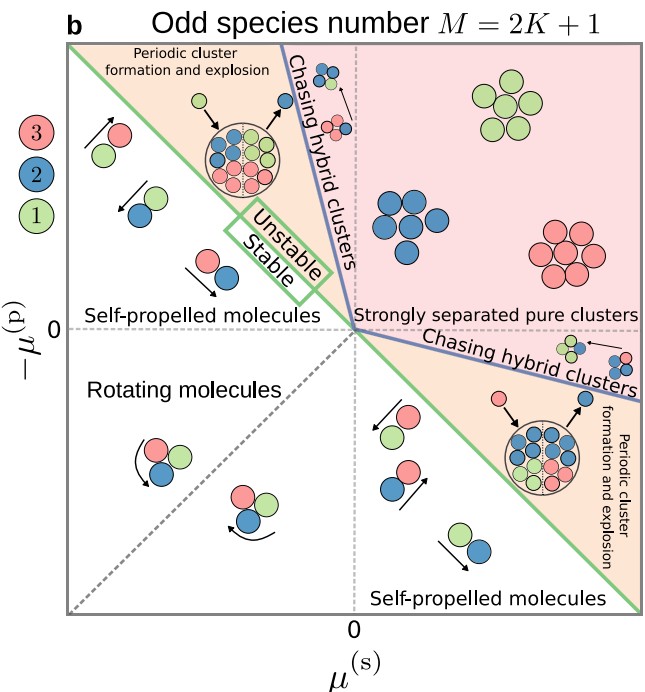

**b**  Odd species number $M = 2K + 1$

**Fig. 2 | Stability diagrams for metabolic cycles.** The cycles contain an even (**a**) or an odd (**b**) number of catalytic species. The interaction motifs in each quadrant of the parameter space are the same as those displayed in Fig. 1d. Details of the behaviour in each phase are given in the text. In both the self-propelled and rotating molecule phases, the molecules exchange particles with one another. Molecule rotation does not occur along the dashed lines corresponding to null imaginary part in the unstable eigenvalue.

i.e. the catalytic species have to be self-attracting for an instability to occur. This is represented in the phase diagrams of Fig. 2: all interaction networks above the green line are unstable. If the condition is not satisfied, the system remains homogeneous, with several possible states: the particles can form transient self-propelled molecules (Supplementary Fig. 3a, Supplementary Movie 1, see Methods for the parameters of all Supplementary Movies), or form more long-lived, rotating molecules (Supplementary Fig. 3b,

Supplementary Movie 2) which exchange particles without growing, as found in particle-based Brownian dynamics simulations of the same system (Methods).

Remarkably, we find key differences between cycles with even or odd number of species. In the case of an even species number $M = 2K$, the eigenvalue with largest real part (which dominates the instability) is real and given by

$$\lambda_K = -2\frac{\alpha\rho_0}{D}\left(\mu^{(p)} - \mu^{(s)}\right), \tag{7}$$

implying that the instability is nonoscillatory with the corresponding eigenvector

$$\delta\rho^K = (1, -1, 1, -1, \cdots, -1). \tag{8}$$

Thus, at onset of instability, all the species with equal parity tend to aggregate together and to separate from the species of opposite parity (Fig. 2a, above the green line). Brownian dynamics simulations show that this prediction carries over to the final phase-separated state; an example is shown in Fig. 3a (Supplementary Movie 3). These simulations show an initial exponential growth of $M$ clusters, each containing all the particles of a given species. The steady state for an even number of self-attracting, cross-repelling species is two large "clusters of clusters", one encompassing clusters of the even-labelled species, the other of the odd-labelled species. Both the transient and the steady state are captured by the growth dynamics shown in Fig. 1e, with the average cluster size initially growing exponentially and saturating at half of the total particle population.

A variety of behaviour is observed in the case with chasing interactions among neighbours, based on the relative values of the chasing strength $|\mu^{(s)} + \mu^{(p)}|$ as compared to the self-attraction strength $|\mu^{(p)} - \mu^{(s)}|$. If both values are of the same order of magnitude, the system behaves similarly to the cross-repelling case, except that the resulting clusters can chase each other or rotate in place (Supplementary Fig. 4a and Supplementary Movie 4). For the cases where the value of the self-attraction is much lower than the chasing strength, fully-hybrid clusters containing all species of the same parity form over longer timescales, as opposed to "clusters of clusters" as in the cross-repelling case (Supplementary Movie 5). Finally, for almost negligible self-attraction, transient oscillations are observed before cluster formation (Supplementary Fig. 4b, Supplementary Movie 6).

For cycles with an odd number of species $M = 2K + 1$, the largest real part corresponds to the complex conjugate pair of eigenvalues (see Supplementary Note 1)

$$\lambda_{K+\frac{1}{2}\pm\frac{1}{2}} = -\frac{\alpha\rho_0}{D}\left(\mu^{(p)} - \mu^{(s)}\right)\left[1 + \cos\left(\frac{\pi}{2K+1}\right)\right]$$
$$\mp i\frac{\alpha\rho_0}{D}\left(\mu^{(s)} + \mu^{(p)}\right)\sin\left(\frac{\pi}{2K+1}\right), \tag{9}$$

suggesting the potential for long-lived oscillations, or even oscillatory steady states, with the real part corresponding to the growth rate of the perturbation and the imaginary part to its oscillation frequency.

The corresponding eigenvectors $\delta\rho^{K+\frac{1}{2}\pm\frac{1}{2}}$ are also a pair of complex conjugates, with components given by

$$\delta\rho_m^{K+\frac{1}{2}\pm\frac{1}{2}} = (-1)^{m-1}\left[\cos\left(\frac{(m-1)\pi}{2K+1}\right) \pm i\sin\left(\frac{(m-1)\pi}{2K+1}\right)\right], \tag{10}$$

for $m = 1, \ldots, 2K + 1$. The species are out of phase by $2\pi/(2K+1)$ with respect to their second-nearest neighbour during the oscillations. Since the number of species is odd, parity-based cluster aggregation is not possible: if two clusters attempt to come together, a third will systematically come to break them apart. For cross-repelling species, this leads to a segregation into single-species clusters which separate

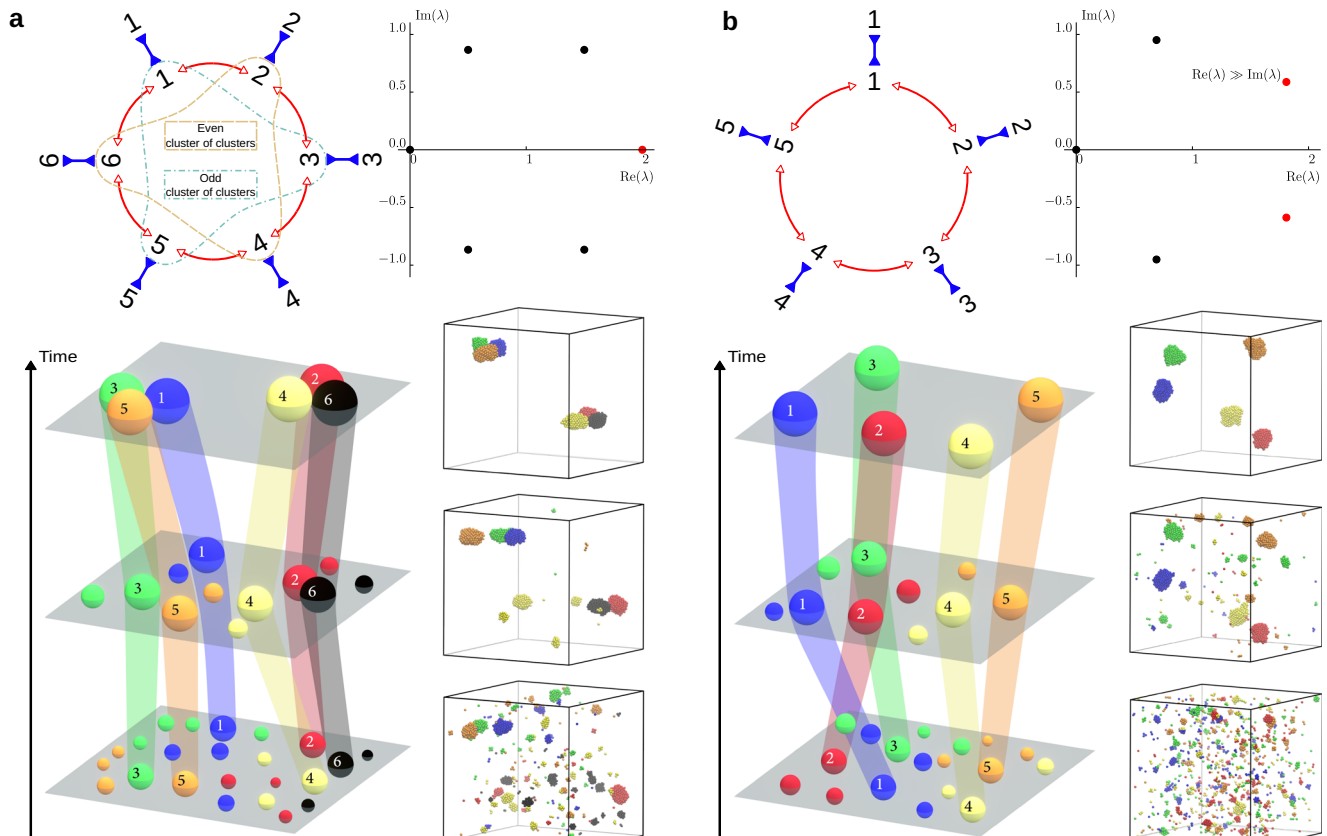

**Fig. 3 | Aggregation dynamics for self-attracting, cross-repelling species.** The cycles contain an even (**a**) or an odd (**b**) number of catalytic species. Top left: interactions between the active species. All species are self-attracting and repel both neighbours in the cycle (corresponding to the top-right quadrants in Fig. 2a, b). Top right: corresponding eigenvalue $\lambda$ spectrum, in units of $-(\mu^{(p)}-\mu^{(s)})\alpha\rho_0/D$ for the real part and $-(\mu^{(s)}+\mu^{(p)})\alpha\rho_0/D$ for the imaginary part. The eigenvalue (or complex conjugate pair) with the largest real part is shown in red. In **b**, the corresponding conjugate pair has an imaginary part much smaller than its real part, so that the dynamics of the system are non-oscillatory. Bottom: Schematic representations of the time evolution of the aggregation dynamics (left) and corresponding snapshots of molecular dynamics simulations (right, see Supplementary Movies 3 and 7 for even and odd cases, respectively) are shown side by side. Dashed lines in **a** indicate the parity-based aggregation that occurs for an even number of species.

in a way that minimizes their overall repulsion (Fig. 3b, Supplementary Movie 7). Similarly to the even case with $M = 2K$, this behavior is captured by the growth statistics displayed in Fig. 1e, where mean cluster size exhibits an initial exponential growth and saturates at a value corresponding to the formation of $M$ individual clusters.

In the case of chasing cross-interactions, oscillations become visible when the growth rate is slower than the oscillation frequency, which corresponds to the condition

$$-\mu^{(p)} \lesssim -\mu^{(s)} \left[ \frac{1 + \cos\left(\frac{\pi}{2K+1}\right) \mp \sin\left(\frac{\pi}{2K+1}\right)}{1 + \cos\left(\frac{\pi}{2K+1}\right) \pm \sin\left(\frac{\pi}{2K+1}\right)} \right], \quad (11)$$

which defines the orange region in Fig. 2b. We note that this inequality only sets an order of magnitude for the transition from oscillatory to non-oscillatory dynamics, rather than a sharp boundary. The behaviour of the system again depends on the relative values of the self attraction magnitude $|\mu^{(p)} - \mu^{(s)}|$ and the chasing strength $|\mu^{(p)} + \mu^{(s)}|$. When self-attraction is weaker than the chasing strength (i.e. close to the instability line), Brownian dynamics simulations indeed show a persistent oscillatory dynamical behaviour with the following choreography for the case in which each species chases after the previous one: a single-species cluster of a species $m$ forms transiently, and is then "invaded" by species $m + 1$, leading to an explosion that disperses species $m$ back into the solution. Species $m$ then invades a cluster of species $m - 1$, and so on, in a sequential order until $M$ explosion events have occurred and the cycle starts again. In the case with $M = 5$ (Fig. 4

and Supplementary Movie 8; see Supplementary Note 2 for a quantification of the oscillation dynamics), we observe that the system comes back to a state similar to the initial one, except for a swap in the locations of the clusters. This change occurs because the clusters of the second-nearest-neighbour species in the cycle tend to form pairs. One component of one of these pairs is replaced in every explosion event by the species preceding it in the cycle, such that, after five explosions, the pairs have been switched in space. The reverse dynamics (species $m$ invading species $m + 1$) are observed if the signs of $\mu^{(s)}$ and $\mu^{(p)}$ are reversed, so that each species chases the next one in the cycle.

For even weaker self-attraction or stronger chasing, the clusters do not have time to form. In this case, oscillations are observed in a dilute mixture of catalytic particles, where clusters are replaced by transient zones of higher concentration (Supplementary Movie 9). This can create a self-stirring solution, favouring the mixing and assembly of solution components in time scales considerably shorter than those allowed by passive diffusion. Lastly, if the perturbation growth rate is instead larger than its oscillation frequency (red region in Fig. 2b), then the dynamics leads to formation of stable clusters. We have observed in simulations the formation of chasing hybrid clusters similar to the case with even number of species (Supplementary Fig. 5, Supplementary Movie 10).

These results can be contrasted with the behaviour of reaction-diffusion systems, which can also undergo instabilities as first formulated by Turing[63], and have been extensively studied for both

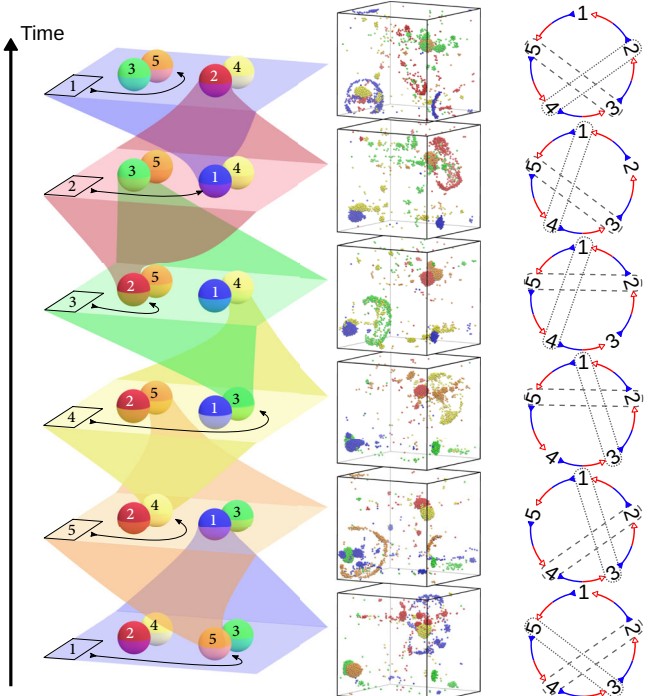

**Fig. 4 | Oscillatory dynamics for an odd number of species with chasing cross-interactions.** A schematic representation of the oscillatory dynamics (left), snapshots of molecular dynamics simulations (middle, see also Supplementary Movie 8), and a diagram of the corresponding species interactions and pairing (right) are shown side by side. Here, dashed and dotted lines represent respectively the pairs on the left and right of the schematic representation. The eigenvalues of the system are as in Fig. 3b, but now $\mathrm{Re}(\lambda) < \mathrm{Im}(\lambda)$ for the most unstable conjugate pair, so that the dynamics of the system are oscillatory.

nonmass-conserving[64–66] and mass-conserving[67,68] reactions. Such systems have been shown to exhibit pattern formation, macroscopic phase separation, or travelling wave fronts. In contrast, the model we study in this work is able to exhibit a larger variety of complex behaviour, because of the non-reciprocal interactions. Additionally, each active species is individually conserved in our model, meaning that it is truly the catalysts and not the reactants that self-organize in this case.

## Discussion

Our work shows that catalytically-active and chemotactic particles participating in a primitive metabolic cycle exhibit a variety of structural complex collective behaviour. Due to the nature of the gradient-mediated interactions involved, such particles are able to interact over large distances, and undergo spontaneous and exponentially rapid cluster formation that serves to support their metabolic function. This feature can help overcome the barrier represented by the time needed for the right types of molecules to meet by chance at sufficiently high concentrations in the first place, and selectively drives the formation of functional metabolic condensates based on the information embedded in the chemical reaction network of the components. This suggests that naturally occurring phoretic transport mechanisms might be able to equip the biological paradigm of liquid-liquid phase separation with an information-controlled strategy for metabolic structure formation. Moreover, since the overall chemical activity of enzymes can be enhanced with suitable clustering behaviour[15,69], the ability to engineer complex clustering features such as those reported here may help improve the design and efficiency of synthetic reaction networks.

Depending on the parity of the number of different species involved in the cycle and on their chemotactic parameters, these clusters might consist of a single or several species, thereby accommodating a range of design strategies for metabolic structure formation. If the number of species in the cycle is odd, chasing interactions may emerge at the macroscopic level, similar to those that have been observed in recent experiments[26,27], although in this case leading to long-lived, system-wide oscillations. Our work suggests that a metabolic cycle consisting of odd number of members may have an advantage (over a cycle with an even number of members) due to the formation of the explosive oscillatory stationary state. It remains to be seen whether the fact that the universal citric acid cycle consists of 11 members can in some way be related to this observation. The observed variety of emergent structural behaviour with highly precise control over the composition of the constituents of the metabolically active clusters hints at a significant possible role for catalytically active molecules at the origin of life: the molecules that are metabolically connected to each other will preferentially and efficiently form active clusters together, hence serving as potential candidates for the nucleation of early forms of life.

What is remarkable about our proposal to use non-reciprocal interactions in this context is that such interactions generically emerge in non-equilibrium systems with chemical catalytic activity[49], which are abundantly present in the cell (molecular catalysts and enzymes involved with metabolic activity) and can be easily synthesized in artificial systems (catalytic colloids, RNA fragments, etc.) for controlled in vitro experiments. In this sense, the theoretical developments that have led to significant progress in the field of active matter in the laboratory setting can now be used to guide new experimental strategies for research in the field of origin of life. Our work offers a theoretical and conceptual platform towards developing this possibility.

## Methods
### Linear stability analysis

We consider a system of $M$ catalytically-active particles described by concentration fields $\rho_m(\mathbf{r}, t)$. A given species $m$ converts its substrate, chemical $(m)$ described by a concentration field $c^{(m)}(\mathbf{r}, t)$, into its product, which will in turn be the substrate $(m+1)$ of the catalytic species $m+1$. This conversion takes place at a rate $\alpha > 0$, which we take to be constant (i.e. catalysis is assumed to take place in the substrate-saturated regime), and equal for all species. The particles are also chemotactic for their substrate and their product, with respective mobilities $\mu^{(s)}$ and $\mu^{(p)}$, again chosen to be equal for all species. We start from the evolution equations for the substrate and product concentrations given in (2). We then consider the effects of a time- and space-dependent perturbation $(\delta\rho_m(\mathbf{r}, t), \delta c^{(m)}(\mathbf{r}, t))$ around an initially homogeneous state $(\rho_0, c_0)$. We also assume a separation of time-scales: as the substrates are typically much smaller than the catalytic particles and thus diffuse faster, we assume that their concentrations equilibrate more quickly to a quasi-steady state for a given configuration of the fields $\rho_m$, meaning that we set $\partial_t c^{(m)} \simeq 0$. Fourier-transforming the linearized equations with respect to space leads to the following equation for the $\delta c^{(m)}$ mode with wavevector $\mathbf{q}$:

$$Dq^2 \delta c^{(m)}(\mathbf{q}, t) = \alpha\big(\delta\rho_{m-1}(\mathbf{q}, t) - \delta\rho_m(\mathbf{q}, t)\big). \tag{12}$$

Reintroducing this perturbation into the linearized equation (1), which we Laplace-transform with respect to time, leads to the system of equations for the different modes with growth rate $\lambda$ and wavevector $\mathbf{q}$:

$$(\lambda + D_\mathrm{p} q^2)\delta\rho_m(\mathbf{q}, \lambda) = -\frac{\alpha\rho_0}{D}\big[\mu^{(s)}\delta\rho_{m-1}(\mathbf{q}, \lambda) \\ + (\mu^{(p)} - \mu^{(s)})\delta\rho_m(\mathbf{q}, \lambda) - \mu^{(p)}\delta\rho_{m+1}(\mathbf{q}, \lambda)\big], \tag{13}$$

which is an eigenvalue equation. It is readily seen that the fastest growing mode is the $\mathbf{q} = 0$ mode. Therefore, we focus on this mode

throughout the paper. The system is unstable when $\mathrm{Re}(\lambda(\boldsymbol{q}=0)) > 0$. Denoting the interaction matrix as $\Lambda_{mn}$ (as defined in Eq. (4)), we obtain the result in Eq. (3).

## Pair interactions between spherical catalytically active particles

In order to perform Brownian dynamics simulations of the system, we calculate the effective interaction between two spherical catalytically active particles in the far-field approximation, which we do in two steps.

We first consider an isolated particle of species $m$, with activity $\alpha$ and radius $R$, taken to be equal for all species. We place the particle at the origin, and use spherical coordinates. The perturbation $\delta c^{(n)}$ induced by the particle, which is assumed to equilibrate quickly with respect to the motion of all particles, is a solution of the Laplace equation:

$$0 = D\nabla^2 \delta c^{(n)}. \tag{14}$$

The corresponding boundary conditions, however, depend on whether the chemical is the substrate ($n = m$), the product ($n = m + 1$), or neither. Indeed, the boundary condition is determined by the diffusive fluxes across the particle surface due to its chemical activity, resulting in

$$-4\pi R^2 D \frac{\partial \delta c^{(n)}}{\partial r}\bigg|_{r=R} = \begin{cases} -\alpha & \text{if } n = m, \\ \alpha & \text{if } n = m+1, \\ 0 & \text{otherwise}. \end{cases} \tag{15}$$

The corresponding solutions for the perturbations are given as

$$\delta c^{(n)}(r) = \begin{cases} -\frac{\alpha}{4\pi D}\frac{1}{r} & \text{if } n = m, \\ \frac{\alpha}{4\pi D}\frac{1}{r} & \text{if } n = m+1, \\ 0 & \text{otherwise}. \end{cases} \tag{16}$$

Now consider a second particle of species $n$ placed at a location $\boldsymbol{r}$. Its velocity $\boldsymbol{v}_{n,m}(\boldsymbol{r})$ in response to the perturbation created by the particle of species $m$ will be

$$\boldsymbol{v}_{n,m}(\boldsymbol{r}) = \begin{cases} -\mu^{(s)}\nabla\delta c^{(n)} & \text{if } n = m+1, \\ -\mu^{(p)}\nabla\delta c^{(n+1)} & \text{if } n = m-1, \\ -\mu^{(s)}\nabla\delta c^{(n)} - \mu^{(p)}\nabla\delta c^{(n+1)} & \text{if } n = m, \\ 0 & \text{otherwise}. \end{cases} \tag{17}$$

Using Eq. (16), the responses can be explicitly written as

$$\boldsymbol{v}_{n,m}(\boldsymbol{r}) = \begin{cases} \frac{\alpha\mu^{(s)}}{4\pi D}\frac{\boldsymbol{r}}{r^3} & \text{if } n = m+1, \\ -\frac{\alpha\mu^{(p)}}{4\pi D}\frac{\boldsymbol{r}}{r^3} & \text{if } n = m-1, \\ \frac{\alpha(\mu^{(p)}-\mu^{(s)})}{4\pi D}\frac{\boldsymbol{r}}{r^3} & \text{if } n = m, \\ 0 & \text{otherwise}, \end{cases} \tag{18}$$

which may be directly compared to the interaction matrix in Eq. (4) of the main text. Note that in general $\boldsymbol{v}_{n,m}(\boldsymbol{r}) \neq -\boldsymbol{v}_{m,n}(-\boldsymbol{r})$ when $n \neq m$, which again highlights the non-reciprocal nature of the interactions.

## Brownian dynamics simulations

We perform Brownian dynamics simulations using a custom program written in the Julia language. We simulate $N$ particles equally distributed among $M$ species. Particles are started out randomly distributed in space, corresponding to a homogeneous state.

**Table 1 | Simulation parameters for the movies referenced in the main text**

| Supplementary Movie number | M | $\tilde{\mu}^{(s)}$ | $\tilde{\mu}^{(p)}$ | dτ | $\tau_{tot}$ |
|---|---|---|---|---|---|
| 1 | 5 | −1.05 | −1 | 0.001 | 2666 |
| 2 | 6 | −0.5 | 1 | 0.0005 | 900 |
| 3 | 6 | 0.5 | −1 | 0.0005 | 900 |
| 4 | 6 | −0.5 | −1 | 0.0005 | 900 |
| 5 | 6 | −0.7 | −0.8 | 0.001 | 8000 |
| 6 | 6 | −0.95 | −1 | 0.001 | 3200 |
| 7 | 5 | 0.5 | −1 | 0.0005 | 900 |
| 8 | 5 | −0.929 | −1.07 | 0.001 | 2000 |
| 9 | 5 | 1.05 | 1 | 0.001 | 2666 |
| 10 | 5 | −0.1 | −0.2 | 0.001 | 6666 |

The equations of motion used in our simulations are

$$\dot{\boldsymbol{r}}_i(t) = \sum_{\substack{j=1 \\ (j\neq i)}}^{N} \boldsymbol{v}_{S(i),S(j)}(\boldsymbol{r}_i - \boldsymbol{r}_j) + \sqrt{2D_{\mathrm{p}}}\,\boldsymbol{\xi}_i, \tag{19}$$

where $i \in \{1, 2, ..., N\}$. Here, $S(i)$ gives the species index corresponding to the particle index $i$, the velocities are calculated using Eq. (18), $D_{\mathrm{p}}$ is the diffusion coefficient of the particles, and $\boldsymbol{\xi}_i$ corresponds to a Gaussian white noise with zero mean and unit variance acting on particle $i$.

The equations of motion are integrated using a forward Euler scheme. At every integration step, an overlap correction is then performed to account for hard-core repulsion between the spheres, using the "elastic collision method"[70]. We simulate the system in a three dimensional box of side length $L$ with periodic boundary conditions, and interactions are treated according to the minimum image convention. Note that we do not use Ewald summation in our numerical simulations, which would be relevant if our goal was to simulate system sizes considerably larger than currently considered in our study.

The particle diameter, $\sigma$, which is taken to be the same for all species, sets the basic length scale of the simulation. We can define reference activity and mobility scales, respectively $\alpha_0$ and $\mu_0$, from which we build a velocity scale $V_0 = \alpha_0\mu_0/(4\pi D\sigma^2)$. From these scales, we can define dimensionless time $\tau = tV_0/\sigma$, activity $\tilde{\alpha} = \alpha/\alpha_0$, and mobility $\tilde{\mu} = \mu/\mu_0$ scales. Finally, we define a reduced particle diffusion coefficient $\tilde{D}_{\mathrm{p}} = D_{\mathrm{p}}/(V_0\sigma)$, which serves as an effective noise intensity or temperature.

## Simulation parameters

All simulations have been performed with a box size $L/\sigma$ chosen such that the total volume fraction of the particles is $\phi = 0.005$, as well as the choice of activity $\tilde{\alpha} = 1$, and noise strength $\tilde{D}_{\mathrm{p}} = 0.02$. Simulations of respectively $M = 5$ ($M = 6$) species are performed with $N/M = 333$ ($N/M = 400$) particles per species. We use the following rule of thumb for parameter choices: the products in the form of $\tilde{\alpha}\tilde{\mu}$ are chosen to be of order unity, while the time step is chosen such that $d\tau \leq 0.001$. The total simulation times are usually of the order of $\tau_{tot} \approx 10^2 - 10^3$. In the Supplementary Information, we describe each simulation movie. For the specific parameters used in each simulation, see Table 1.

## Cluster growth law determination

In order to qualitatively determine the cluster growth law for both an even (see Fig. 3a of the main text and Supplementary Movie 7) and an odd (see Fig. 3b of the main text and Supplementary Movie 3) number of species, we use a simple clustering algorithm implemented in the Julia programming language: starting from each individual particle regarded as a cluster of size 1, we assign two particles to be in the same cluster if their distance is below a threshold, which we choose to be 1.1

times the particle diameter $\sigma$. In order to speed up the calculations, we additionally use a link-list algorithm with a cell size $1.1\sigma$, which only requires calculation of the distance of each particle to the particles in its vicinity. We run 100 simulations using the parameters given in the previous subsection for Supplementary Movies 3 and 7, with a longer total time $\tau_{tot} = 400$. We then perform an ensemble average on these data to obtain the growth laws shown in Fig. 1e.

## Data availability

The data supporting the main findings of this study are available in the paper and its Supplementary Information. Any additional data can be made available upon request.

## Code availability

The algorithms for the codes supporting the main findings of this study are available in the paper and its Supplementary Information. Any additional information concerning the code can be made available upon request.

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

## Acknowledgements

We acknowledge support from the Max Planck School Matter to Life and the MaxSynBio Consortium which are jointly funded by the Federal Ministry of Education and Research (BMBF) of Germany and the Max Planck Society.

## Author contributions

V.O-.R., J.A-.C., and R.G. designed the research, conducted the research, analyzed the data, and wrote the paper.

## Funding

## Competing interests

The authors declare no competing interests.
