## [Peer Review File · Nature Communications]

REVIEWER COMMENTS

Reviewer #1 (Remarks to the Author):

The manuscript considers possible explanations for the emergence of a particular nonequilibrium phenomenon (life) from a certain type of nonequilibrium dynamics (nonreciprocal interactions). More specifically, it studies the phase behavior of a model catalytic cycle.

Non-reciprocal interactions are a very timely topic in active matter physics. The results are exciting and novel as far as I can tell, and I expect them to stimulate further research in the active matter community. I would therefore recommend the article for publication in "Nature Communications" after the authors have addressed the following concerns:

1) The central claim of the article is that non-reciprocal interactions are relevant for explaining the origin of life. What is actually demonstrated is that a system of reaction-diffusion equations that is taken to represent a catalytic cycle exhibits a certain phase behavior. While this is certainly interesting, validating the central claim of the article requires (in my opinion) that the connection of these results to biology/chemistry is made a bit clearer. In what sense does the model resemble things that one would expect to find in a primordial soup, and in what sense does the behavior of the model constitute something like 'life'?

2) Also, do the specific observations made in the article (such as the differences between systems with even and odd species numbers or the chasing cross-interactions) have a specific relevance in this biological context? For example, would they suggest that catalytic cycles with an even/odd number of "participants" are the most likely origin of life?

3) It is moreover claimed in the abstract that the results "shed light on possibilities that may be explored in designing efficient synthetic cycles", but this is not really explained anywhere in the article. The authors should either discuss this possibility in more detail in the article (how precisely can these effects be exploited in synthetic cycles?) or remove the claim.

4) The article has 38 references, 13 of which are self-citations. I think the authors can discuss a bit more work also from other authors, for example

<https://doi.org/10.1038/s41467-022-30834-2>

<https://doi.org/10.1039/D2CP02542F>

5) The properties of the proposed model are compared to other active matter phenomena (active phase separation, Keller-Segel model), but not so much to existing results on pattern formation in reaction-diffusion systems, which seem to be even more relevant here. How do the effects observed here fit into the general theory of pattern formation in reaction-diffusion systems (Turing/Hopf instabilities)?

Reviewer #2 (Remarks to the Author):

Title: Self-organization of primitive metabolic cycles due to non-reciprocal interactions

Authors: Vincent Ouazan-Reboul, Jaime Agudo-Canalejo, and Ramin Golestanian

In this paper, authors demonstrated that cascade chemical reaction networks catalyzed by chemo-phoretic particles generate various dynamical structures depending on their chemo-phoretic mobility and number of catalytic species, which includes self-propelled molecules, rotating molecules, and aggregates depending on the parity. The observed complex collective behavior should be of interest to the active matter community. My main concern with this paper is that it is difficult to understand how the collective behavior reported in the paper is related to primitive metabolic cycles in the primordial soup.

1) My first impression of this manuscript is that the authors make little mention of previous works on how the first metabolic networks emerged at the origin of life. On the emergence of primitive metabolic pathways, there have been many studies, which started from the idea of autocatalytic set by Kauffman [Kauffman, S.A., *J. Theor. Biol.* 119, 1 (1986.)], and developed to more rigorous model of RAF theory [e.g., Steel, M. *Appl. Math. Lett.* 3, 91 (2000), Xavier J.C., et al., *Proc. R. Soc. B* 287: 20192377 (2020)]. These theoretical models have been examined experimentally [e.g., Vaidya, N., et al., *Nature* 491, 72 (2012)]. The author should refer to previous studies on primitive metabolic systems and clarify the position of the present study with respect to those studies.

2) The authors state "The observed variety of emergent structural behavior with highly precise control over the composition of the constituents of the metabolically active clusters hints at a significant possible role for catalytically active molecules at the origin of life" in the conclusion section. I cannot understand why the various structures in the phase diagram (Fig. 2) play such an important role in the emergence of the catalytic network at the origin of life. Do these structures promote the formation of sustainable metabolic reaction systems? Do they improve the efficiency of the production of final products in metabolic reaction systems? The relationship between the observed complex collective behavior and non-equilibrium conditions necessary for the formation of primitive metabolic structures should be clearly explained.

Minor points

3) What colloidal particles present in primordial soups are candidates as chemically-active particles with chemo-phoretic effects? Experimentally, various RNA molecules all help each other's formation from their basic building blocks, in a network of molecular collaboration [e.g., Horning, D.P., Joyce, G.F., *PNAS* 113, 9786 (2016), Higgs, P.G., Lehman, N., *Nat. Rev. Genet.* 16, 7 (2015), Nghe, P., et al., *Mol. BioSyst.* 11, 3206 (2015), Vaidya, N., et al., *Nature* 491, 72 (2012)]. Do the authors suggest that chemically-active particles with chemo-phoretic effects play an important role in such a metabolic cycle? I think it is important to indicate what metabolic cycles at the origin of life the authors are envisioning.

4) The authors assume that all catalyst species have the same parameters α , $\mu(s)$, and $\mu(p)$, and are present in the system at identical initial concentrations. This assumption does not seem realistic in primordial soup. If this assumption is violated, would it have a significant impact on your conclusions?

I think this is an excellent paper in the active matter field. However, in order to attract interest not only from the field of active matter but also from more researchers interested in the origin of life, it is desirable to clarify the points mentioned above and lower the barriers between the fields.

Self-organization of primitive metabolic cycles due to non-reciprocal interactions – Point-by-point response to the reviewers' comments

(Dated: May 22, 2023)

This document contains our detailed answers to the comments of the reviewers. In the updated and marked version, blue pieces of text correspond to pieces added or edited in response to the reviewers comments. Below, the comments of the referees are displayed in *italic*, our answers are in standard roman font.

Both referees were positive in their evaluation of the manuscript. Reviewer #1 finds that “The results are exciting and novel as far as [they] can tell”, “expect[s] them to stimulate further research in the active matter community”, and “would therefore recommend the article for publication in Nature Communications”. Reviewer # 2 considers that “The observed complex collective behavior should be of interest to the active matter community”, and “think[s] this is an excellent paper in the active matter field”.

The referees had many helpful recommendations and queries concerning the link between our work and the relevant chemistry and biology that connects it to the origin of life discussions, as well as other clarification points. We thank the reviewers for raising these points, which we hope to have satisfactorily addressed in the revised manuscript, and which have helped make the manuscript more complete.

Below are our point-by-point answers to the comments of the referees.

REVIEWER #1

1. *“The central claim of the article is that non-reciprocal interactions are relevant for explaining the origin of life. What is actually demonstrated is that a system of reaction-diffusion equations that is taken to represent a catalytic cycle exhibits a certain phase behavior. While this is certainly interesting, validating the central claim of the article requires (in my opinion) that the connection of these results to biology/chemistry is made a bit clearer. In what sense does the model resemble things that one would expect to find in a primordial soup, and in what sense does the behavior of the model constitute something like ‘life’?”*

The field of origin of life includes many plausible scenarios for the different stages that may have existed before the formation of the current form. Every proposed scenario has strengths and weaknesses and a comprehensive dialogue surrounding these points is the body that currently forms this scientific field. Our contribution engages a number of specific issues in connection with some of these scenarios and offers a potential resolution for some of the critical issues with the introduction of a new paradigm from non-equilibrium statistical physics of chemically active matter, namely, non-reciprocal interactions.

We particularly highlight three general points: (i) the so-called “chicken-and-egg” problem concerning whether information came first and structure was developed based on it or structure came first and information was developed around it, (ii) the so-called containment problem, namely how the right ingredients to form the right structures accumulated at sufficiently high densities in the first place to provide the chance of encounter for these active ingredients, and (iii) the physical question of time-scale needed for the evolution of the current form of life from the prebiotic soup. Our proposed paradigm takes a specific stance concerning these questions and makes specific quantitative predictions that will help to make progress towards resolving them. Super-exponential aggregation offers a solution to the containment problem and the slowness problem. We find evidence that suggests that information related to a metabolic cycle can be the organizer of the metabolic cycle in space and time, via non-reciprocal interactions that exist in these systems. We have now expanded the manuscript to make a more explicit connection with the existing discussions in the field of origins of life, and included discussions of the points mentioned above.

2. *“Also, do the specific observations made in the article (such as the differences between systems with even and odd species numbers or the chasing cross-interactions) have a specific relevance in this biological context? For example, would they suggest that catalytic cycles with an even/odd number of “participants” are the most likely origin of life?”*

The general idea proposed in our work is that the introduction of non-reciprocal interactions will have many interesting and non-trivial consequences that can contribute to many of the ongoing discussions around origin of life and provide scenarios for spontaneous self-organization of metabolic cycles.

One of the predictions of our theory is the possibility that prebiotic metabolic cycles with an odd number of catalysts might have had a selective advantage due to the existence of a stationary state cyclic mode with explosive behaviour. We have now highlighted the citric acid cycle that involves 11 catalysts, and speculated

that the odd number of catalysts in this so-called universal cycle might be an interesting point to investigate in search of a possible connection with our theory.

3. *“It is moreover claimed in the abstract that the results “shed light on possibilities that may be explored in designing efficient synthetic cycles”, but this is not really explained anywhere in the article. The authors should either discuss this possibility in more detail in the article (how precisely can these effects be exploited in synthetic cycles?) or remove the claim.”*

We believe that the statement is justified because engineering a desired emergent property will always be helped by studies that show how tuning microscopic parameters can lead to different types of self-organization. Moreover, there exist many studies that highlight how clustering leads to enzymatic activity with higher efficiency. Therefore, the ability to control clustering of enzymes will likely be useful in designing applications. We have now added citations to this group of studies to support more immediate application of our studies.

4. *“The article has 38 references, 13 of which are self-citations. I think the authors can discuss a bit more work also from other authors, for example <https://doi.org/10.1038/s41467-022-30834-2> <https://doi.org/10.1039/D2CP02542F>”*

We would like to thank the reviewer for suggesting these references, which we have now added to the bibliography of the manuscript together with many more (we now have 67 refs), as we have significantly expanded the discussions in connection with other works. The two references proposed here concern physical processes (other than non-reciprocal interactions) that might be relevant for origin of life discussions.

5. *“The properties of the proposed model are compared to other active matter phenomena (active phase separation, Keller-Segel model), but not so much to existing results on pattern formation in reaction-diffusion systems, which seem to be even more relevant here. How do the effects observed here fit into the general theory of pattern formation in reaction-diffusion systems (Turing/Hopf instabilities)? ”*

We have added a discussion concerning reaction-diffusion models and the relevant comparison. We would like to thank the reviewer for suggesting this connection, which further highlights the unique features that emerge from non-reciprocal interactions in chemically active systems.

REVIEWER #2

1. *“My first impression of this manuscript is that the authors make little mention of previous works on how the first metabolic networks emerged at the origin of life. On the emergence of primitive metabolic pathways, there have been many studies, which started from the idea of autocatalytic set by Kauffman [Kauffman, S.A., *J. Theor. Biol.* 119, 1 (1986.)], and developed to more rigorous model of RAF theory [e.g., Steel, M. *Appl. Math. Lett.* 3, 91 (2000), Xavier J.C., et al., *Proc. R. Soc. B* 287: 20192377 (2020)]. These theoretical models have been examined experimentally [e.g., Vaidya, N., et al., *Nature* 491, 72 (2012)]. The author should refer to previous studies on primitive metabolic systems and clarify the position of the present study with respect to those studies.”*

We would like to thank the reviewer for suggesting these references, which we have now added to the bibliography of the manuscript together with many more, as we have significantly expanded the relevant discussions in connection with other works.

2. *“The authors state “The observed variety of emergent structural behavior with highly precise control over the composition of the constituents of the metabolically active clusters hints at a significant possible role for catalytically active molecules at the origin of life” in the conclusion section. I cannot understand why the various structures in the phase diagram (Fig. 2) play such an important role in the emergence of the catalytic network at the origin of life. Do these structures promote the formation of sustainable metabolic reaction systems? Do they improve the efficiency of the production of final products in metabolic reaction systems? The relationship between the observed complex collective behavior and non-equilibrium conditions necessary for the formation of primitive metabolic structures should be clearly explained.”*

The field of origin of life includes many plausible scenarios for the different stages that may have existed before the formation of the current form. Every proposed scenario has strengths and weaknesses and a comprehensive dialogue surrounding these points is the body that currently forms this scientific field. Our contribution engages a number of specific issues in connection with some of these scenarios and offers a potential resolution for some of the critical issues with the introduction of a new paradigm from non-equilibrium statistical physics of chemically active matter, namely, non-reciprocal interactions.

We particularly highlight three general points: (i) the so-called “chicken-and-egg” problem concerning whether information came first and structure was developed based on it or structure came first and information was developed around it, (ii) the so-called containment problem, namely how the right ingredients to form the right structures accumulated at sufficiently high densities in the first place to provide the chance of encounter for these active ingredients, and (iii) the physical question of time-scale needed for the evolution of the current form of life from the prebiotic soup. Our proposed paradigm takes a specific stance concerning these questions and makes specific quantitative predictions that will help to make progress towards resolving them. Super-exponential aggregation offers a solution to the containment problem and the slowness problem. We find evidence that suggests that information related to a metabolic cycle can be the organizer of the metabolic cycle in space and time, via non-reciprocal interactions that exist in these systems. We have now expanded the manuscript to make more explicit connection with the existing discussions in the field of origins of life, and included discussions of the points mentioned above.

The general idea proposed in our work is that the introduction of non-reciprocal interactions will have many interesting and non-trivial consequences that can contribute to many of the ongoing discussions around origin of life and provide scenarios for spontaneous self-organization of metabolic cycles. This feature can selectively drive the formation of functional metabolic condensates based on the information embedded in the chemical reaction network of the components. This suggests that naturally occurring phoretic transport mechanisms might be able to equip the biological paradigm of liquid-liquid phase separation with an information-controlled strategy for metabolic structure formation.

One of the predictions of our theory is the possibility that prebiotic metabolic cycles with an odd number of catalysts might have had a selective advantage due to the existence of a stationary state cyclic mode with explosive behaviour. We have now highlighted the citric acid cycle that involves 11 catalysts, and speculated that the odd number of catalysts in this so-called universal cycle might be an interesting point to investigate in search of a possible connection with our theory.

We have now added extensive discussions on these points to the revised manuscript.

3. *“What colloidal particles present in primordial soups are candidates as chemically-active particles with chemophoretic effects? Experimentally, various RNA molecules all help each other’s formation from their basic building blocks, in a network of molecular collaboration [e.g., Horning, D.P., Joyce, G.F., PNAS 113, 9786 (2016), Higgs, P.G., Lehman, N., Nat. Rev. Genet. 16, 7 (2015), Nghe, P., et al., Mol. BioSyst. 11, 3206 (2015), Vaidya, N., et al., Nature 491, 72 (2012)]. Do the authors suggest that chemically-active particles with chemophoretic effects play an important role in such a metabolic cycle? I think it is important to indicate what metabolic cycles at the origin of life the authors are envisioning.”*

The effect is present as soon as a particle exhibits catalytic activity, independently of its detailed chemical characteristics. Therefore it should be relevant to nucleic acids as well as other catalysts and enzymes. Indeed, it is the generic aspect of this phenomenon that makes it an important player in the organization of metabolic cycles in our opinion.

Importantly, it is not necessary to have colloidal particles that are of larger scales (microns or so) to experience this effect. The larger scale colloidal particles are merely useful in bridging the scales for our observations, as we cannot directly observe molecules. We have now clarified this point in the manuscript. Moreover, we have added the references concerning the RNA literature in the discussions of our manuscript.

4. *“The authors assume that all catalyst species have the same parameters α , $\mu^{(s)}$, and $\mu^{(p)}$, and are present in the system at identical initial concentrations. This assumption does not seem realistic in primordial soup. If this assumption is violated, would it have a significant impact on your conclusions?”*

This is indeed a very interesting question. We have shown in other publications that a full classification of different types of behaviour is possible in such systems, with each class pertaining to a large part of the parameter space. Therefore, we expect the behaviour to be relatively robust with respect to changes in the parameters. We have now added a discussion on this question, with the relevant reference to support the statement.

REVIEWER COMMENTS

Reviewer #1 (Remarks to the Author):

The revised version of the manuscript addresses all my concerns, and I believe that the additions made by the authors make clearer what the physical and biological significance of the results is. I recommend the article for publication in "Nature Communications".

Reviewer #2 (Remarks to the Author):

In this revision, the authors addressed most of my questions and comments. My remaining question is concerning the previous comment 3: "What colloidal particles present in primordial soups are candidates as chemically-active particles with chemophoretic effects?". I well understand your response "The effect is present as soon as a particle exhibits catalytic activity, independently of its detailed chemical characteristics. Therefore it should be relevant to nucleic acids as well as other catalysts and enzymes.". However, to my knowledge, I am unaware of any examples of nucleic acids or enzymes which show chemophoretic effects. Please show (or discuss) whether the nucleic acids and catalysts relevant to the origin of life can satisfy the quantitative $\mu(s)$ and $\mu(p)$ conditions necessary to observe collective motion demonstrated in this paper.

Self-organization of primitive metabolic cycles due to non-reciprocal interactions – Point-by-point response to the reviewers' comments

(Dated: June 9, 2023)

This document contains our detailed answers to the comments of Reviewer #2. In the updated and marked version, blue pieces of text correspond to pieces added or edited in response to the reviewers comments. Below, the comments of the referees are displayed in *italic*, our answers are in standard roman font.

REVIEWER #2

1. *In this revision, the authors addressed most of my questions and comments. My remaining question is concerning the previous comment 3: “What colloidal particles present in primordial soups are candidates as chemically-active particles with chemophoretic effects?”. I well understand your response “The effect is present as soon as a particle exhibits catalytic activity, independently of its detailed chemical characteristics. Therefore it should be relevant to nucleic acids as well as other catalysts and enzymes.”. However, to my knowledge, I am unaware of any examples of nucleic acids or enzymes which show chemophoretic effects. Please show (or discuss) whether the nucleic acids and catalysts relevant to the origin of life can satisfy the quantitative $\mu(s)$ and $\mu(p)$ conditions necessary to observe collective motion demonstrated in this paper.*

There have been a number of experimental reports of nucleic acids and enzymes that show chemophoretic effects, broadly understood here as biased motion in the presence of gradients of a chemical (typically a substrate or a functionally related molecule) that cannot be explained by undirected diffusion alone. Perhaps of particular relevance in the prebiotic context are those involving nucleic acids:

i) In [Yu et al. (2009). Molecular propulsion: chemical sensing and chemotaxis of DNA driven by RNA polymerase. *Journal of the American Chemical Society*, 131(16), 5722-5723.], biased motion of a molecular complex consisting of a DNA template and associating RNA polymerases was observed in gradients of transcription substrates (nucleoside triphosphates).

ii) In [Ramm et al. (2021). A diffusiphoretic mechanism for ATP-driven transport without motor proteins. *Nature Physics*, 17(7), 850-858.], DNA origami structures were shown to migrate in response to gradients of proteins belonging to the MinDE system of *E. Coli*.

In a number of other works (reviewed in [Agudo-Canalejo et al. (2018). Enhanced diffusion and chemotaxis at the nanoscale. *Accounts of chemical research*, 51(10), 2365-2372.]), biological enzymes have been observed to show biased motion in the presence of their chemical substrate:

i) [Sengupta et al. (2013). Enzyme molecules as nanomotors. *Journal of the American Chemical Society*, 135(4), 1406-1414.] showed chemotaxis of urease, catalase, and glucose oxidase towards their substrates

ii) [Sengupta et al. (2014). DNA polymerase as a molecular motor and pump. *ACS nano*, 8(3), 2410-2418.] showed chemotaxis of DNA polymerase towards nucleotides and cofactors

iii) [Dey et al. (2014). Chemotactic separation of enzymes. *ACS nano*, 8(12), 11941-11949.] showed chemotaxis of catalase, urease and β -galactosidase towards their substrates

iv) [Zhao, X et al. (2018). Substrate-driven chemotactic assembly in an enzyme cascade. *Nature Chemistry*, 10(3), 311-317.] showed chemotaxis of hexokinase, phosphoglucose isomerase, phosphofructokinase and aldolase, all of which participate in the glycolysis cascade, in response to their respective substrates.

v) [Jee et al. (2018). Enzyme leaps fuel antichemotaxis. *Proceedings of the National Academy of Sciences*, 115(1), 14-18.] and [Jee et al. (2018). Catalytic enzymes are active matter. *Proceedings of the National Academy of Sciences*, 115(46), E10812-E10821.] showed chemotaxis of urease and acetylcholinesterase in response to substrate gradients

Lastly, [Wang et al. (2020). Boosted molecular mobility during common chemical reactions. *Science*, 369(6503), 537-541.] reported chemotaxis of molecular-scale catalysts in the presence of gradients of their chemical substrate, in this case for the copper-catalyzed click reaction and the Diels-Alder reaction.

As the reviewer can see, there are many examples of such chemophoretic motion, even of nucleic acids that are particularly relevant in a RNA world prebiotic context. As we have already stressed, phoresis is a generic mechanism arising from the physical characteristics of the object in question, so we think it is plausible that other nucleic acids undergo phoresis, and it is important to think about its potential role in an origin of life

scenario. At this stage, however, there isn't sufficient information about the μ 's for specific catalysts in specific metabolic cycles relevant to pre-biotic chemistry for us to make quantitative claims about these values.

In the revised version, we now cite and discuss a representative sample of the references just mentioned, as well as our review on enzyme chemotaxis, on page 3.

REVIEWERS' COMMENTS

Reviewer #2 (Remarks to the Author):

The added literatures indicate that prebiotic molecules may have chemophoretic nature, which strengthens the message of this paper. I recommend this article for publication in "Nature Communications".